# Identifying triplet pathways in dilute pentacene films

Daphné Lubert-Perquel[1], Enrico Salvadori [2,3], Matthew Dyson[4,5], Paul N. Stavrinou[6], Riccardo Montis[1] Hiroki Nagashima [7], Yasuhiro Kobori [7,8], Sandrine Heutz [1,9] & Christopher W.M. Kay [3,10]

Building efficient triplet-harvesting layers for photovoltaic applications requires a deep understanding of the microscopic properties of the components involved and their dynamics. Singlet fission is a particularly appealing mechanism as it generates two excitons from a single photon. However, the pathways of the coupled triplets into free species, and their dependence on the intermolecular geometry, has not been fully explored. In this work, we produce highly ordered dilute pentacene films with distinct parallel and herringbone dimers and aggregates. Using electron paramagnetic resonance spectroscopy, we provide compelling evidence for the formation of distinct quintet excitons in ambient conditions, with intrinsically distinctive electronic and kinetic properties. We find that the ability of quintets to separate into free triplets is promoted in the parallel dimers and this provides molecular design rules to control the triplets, favouring either enhanced photovoltaic efficiency (parallel) or strongly bound pairs that could be exploited for logic applications (herringbone).

[1] London Centre for Nanotechnology and Department of Materials, Imperial College London, Prince Consort Road, London SW7 2BP, UK. [2] School of Biological and Chemical Science, Queen Mary University of London, Mile End Road, London E1 4NS, UK. [3] Institute of Structural and Molecular Biology and London Centre for Nanotechnology, University College London, Gower Street, London WC1E 6BT, UK. [4] Department of Physics and Centre for Plastic Electronics, Imperial College London, London SW7 2AZ, UK. [5] Molecular Materials and Nanosystems and Institute for Complex Molecular Systems, Eindhoven University of Technology, P.O. Box 513, 5600 MB Eindhoven, The Netherlands. [6] Department of Engineering Science, Oxford University, Parks Road, Oxford OX1 3PJ, UK. [7] Molecular Photoscience Research Center, Kobe University, 1-1 Rokkodaicho, Nada-ku Kobe 657-8501, Japan. [8] Department of Chemistry, Graduate School of Science, Kobe University, 1-1 Rokkodaicho, Nada-ku, Kobe 657-8501, Japan. [9] Centre for Plastic Electronics, Imperial College London, London SW7 2AZ, UK. [10] Department of Chemistry, University of Saarland, 66123 Saarbrücken, Germany. Correspondence and requests for materials should be addressed to E.S. (email: e.salvadori@qmul.ac.uk) or to S.H. (email: s.heutz@imperial.ac.uk) or to C.W.M.K. (email: c.kay@ucl.ac.uk)

nterest in the photophysics of singlet fission (SF) has dramatically increased in recent years due to the possibility of overcoming thermodynamic limitations in the efficiencies of organic electronic, organic spintronic and hybrid organic/inorganic structures[1–3]. SF is the photophysical process by which a single photon absorbed by a pair of interacting chromophores generates two triplet excitons. Initially these are strongly coupled triplets but they may dissociate into free triplets or recombine via triplet–triplet annihilation. In its basic description, SF requires that the energy of the incoming photon is more than double that of the generated triplets. Being a spin-allowed process, it can be fast enough to out-compete prompt fluorescence. Harnessing this mechanism holds the promise to exceed the Shockley–Queisser limit[4], and quantum efficiencies >100% have been reported[5,6]. Only a few molecules are known to undergo SF, of which polyacenes are the most extensively investigated[7–12]. Recent studies identify the bound nature of coupled triplets[11] and show evidence of the quintet nature of coupled triplets in isotropic frozen solutions and amorphous films[12,13]. However, much remains unknown as to the precise nature of the multiexciton (paramagnetic) states and how the coupling between chromophores depends upon their geometry.

To address this, efforts have been devoted to the investigation of covalently bound chromophore dimers and van der Waals aggregates[14–16]. These provide excellent tunability of the electronic coupling but are impractical for real devices, as they are difficult to synthesise or are stable only under strictly controlled conditions. Spin-coating of pristine TIPS-tetracene and TIPS-pentacene provides a convenient medium to investigate SF mechanisms[12,13,17,18], but these disordered films require ultra-fast optics and/or cryogenic temperatures to study the population of spin states in order to suppress fast exciton diffusion[2]. Extensive theoretical and experimental work has been devoted to the description of singlet exciton properties of molecular aggregates, films and crystals, in terms of H- and J-type aggregates—however, a recent study highlighted that charge transfer (CT) could introduce states beyond this traditional description, particularly for the herringbone structure adopted by solid-state polyacenes[15,19]. While it has been shown that the configuration of pentacene dimers directly impacts on the SF pathways due to differing signs of the singlet exciton–exciton coupling[20], analogous descriptions are lacking for the corresponding triplet excitons. As would be expected, local microstructure plays an important role in the dynamics of SF in pentacene compounds[8,10,21,22], as the physical arrangement of molecules in a solid determines the properties of excitons, including the extent of delocalisation and associated relative admixture of CT configurations in the description of the exciton wave function[23]. Moreover, it should be noted that the vast majority of SF studies are based on transient absorption spectroscopy due to its fs/ps time resolution at room temperature. Recently, electron paramagnetic resonance (EPR) spectroscopy at cryogenic temperatures was used to selectively address triplet and quintet states that can be observed, but not distinguished, in optical/transient absorption experiments[12,13]. This approach, which not only allows the identification of the triplet pathways as a function of aggregation, also does so according to their molecular configurations.

To enable detection and characterisation of the high-spin intermediates formed upon SF, we investigated ordered films of pentacene in a p-terphenyl matrix at different concentrations. Pentacene (Fig. 1a) is selected as the benchmark molecule for SF, whilst p-terphenyl (Fig. 1b) provides a well-defined host. The crystal structure of the pentacene in p-terphenyl is documented—pentacene substitutes into one of the two inequivalent sites, due to the herringbone structure of the host lattice[24–26]. As assessed by optical spectroscopy, dilution is shown to have little effect on the efficiency of SF[19]; however, in combination with oriented samples and EPR spectroscopy, it allows the kinetics of distinct spin species to be derived. Furthermore, as p-terphenyl is not a conjugated system, CT between host and dopant is suppressed.

Figure 1c depicts the energy levels and kinetics of the excited states in pentacene. Triplet states are populated via two distinct mechanisms that occur on different timescales and involve one or two pentacene units. SF (teal) is a CT-mediated decay from $S_1$ to $^n(TT)$, where n represents the spin species, requiring two interacting chromophores. These initial $^n(TT)$ states either dissociate into free triplets with uses in organic photovoltaic

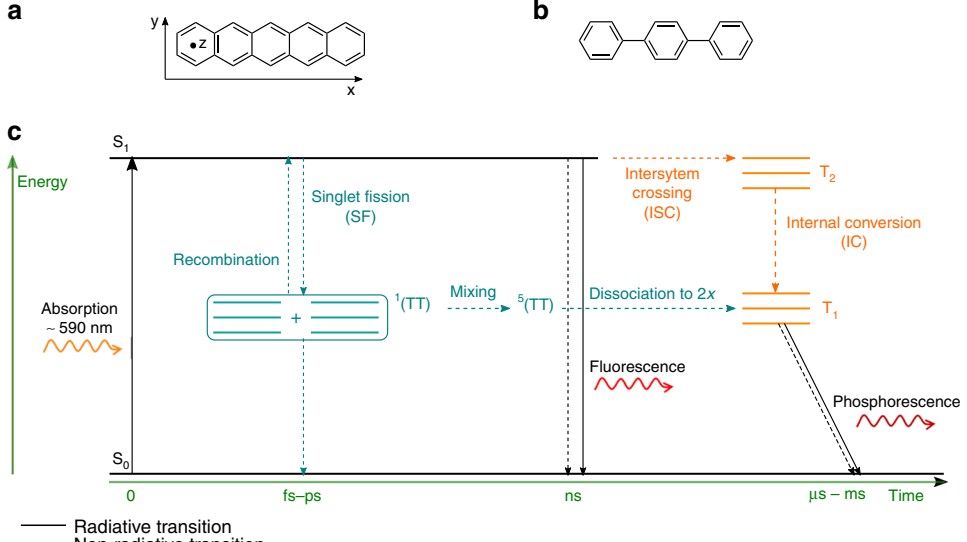

**Fig. 1** Molecular structures and Jablonski diagram with timescale. **a** Molecular structure of pentacene with corresponding ZFS axes. **b** Molecular structure of p-terphenyl. **c** As a result of photoexcitation, pentacene generates triplets via two distinct mechanisms. Coupled pentacene molecules undergo fast (fs-ps) SF leading to two strongly coupled $^1(TT)$ states which can then populate the quintet state (teal). These can recombine or ultimately dissociate into free triplets. Alternatively, individual pentacene molecules can transition to triplet states via ISC (orange), which is energetically less favourable due the required spin-flip and therefore occurs on slower timescales (ns)

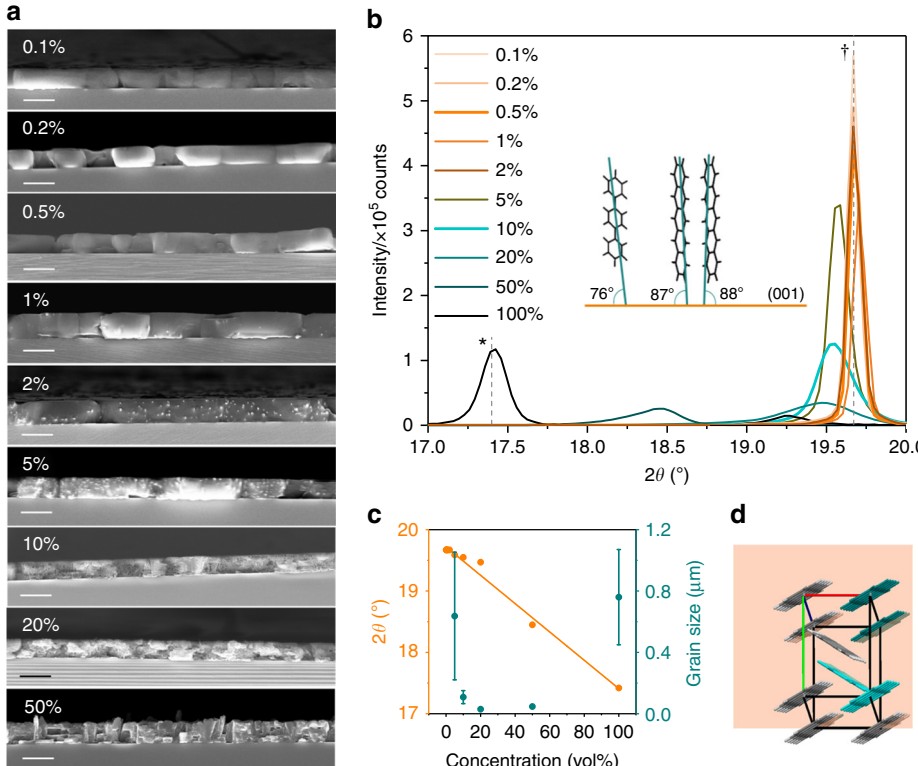

**Fig. 2** Morphology and structure of the 1-μm-thick pentacene films as a function of concentration. **a** SEM cross-sections with a 1-μm scale bar. The dilute samples form large platelets typical of *p*-terphenyl, whereas an increase in dopant concentration leads to fragmentation into smaller faceted grains. **b** Close-up of the (003) peak where (\*) corresponds to the pentacene peak position and (†) *p*-terphenyl position. The inset shows the contact angle of *p*-terphenyl (left) and pentacene (right) on the substrate. **c** Variation in scattering angle and grain size (calculated from the Scherrer equation [31]) with increasing pentacene concentration, indicating a linear shift to towards lower scattering angles (i.e. increased lattice spacing); 5 and 100% pentacene are close to instrument resolution and consequently have the largest error in grain size. All films are 1 μm except the 100% pentacene which is 100 nm—XRD intensity is normalised to the thickness. The error in crystallite size was estimated as the difference between the Lorentzian and Gaussian calculation of the Scherrer equation. [47] **d** Unit cell of the pentacene showing the parallel dimer (teal) and herringbone dimer (cyan)

devices[1,2,5,27–29] or recombine into an emissive $S_1$ state. Alternatively, intersystem crossing (ISC) (orange) generates triplets via an $S_1$-$T_2$-$T_1$ transition. This is a single chromophore process, which is slower (nanosecond timescale) due to the spin-flip requirement, yet efficient in pentacene due to the quasi-resonant nature of the $S_1$ and $T_2$ states[30]. It follows that manipulating chromophore concentration allows control of the pathway in Fig. 1 even under ambient conditions. EPR spectroscopy is used to characterise the progression from isolated to coupled photo-induced spin states in ordered films, and hence, the effect of molecular aggregation and orientation on the triplet pathways can be elucidated. Crucially, the conditions chosen for the experiments are those experienced by a real-life device, i.e. room temperature and in air. To achieve these goals, films are prepared by organic molecular beam deposition. The rationale behind this is twofold: (i) as co-evaporation is kinetically not thermo-dynamically driven, this technique enables pentacene to exceed its solubility limit and results in homogeneously doped films with no phase separation yet with defined structural ordering; (ii) precise control of pentacene concentration allows the progression from isolated molecules to van der Waals dimers and clusters to be studied. These aspects are significant as they result in localised long-lived excitons that can be studied under ambient conditions, a prerequisite for any functional material designed for devices.

In this work, we report the effect of two distinct molecular geometries within aggregates, parallel and herringbone, on SF. We are able to distinguish both triplets and quintets generated at room temperature and provide a kinetic model for their

dynamics. We find that quintets are long lived even at room temperature and that the parallel geometry is favourable for triplet dissociation and therefore charge generation in organic semiconductor devices.

## Results

**Structural characterisation.** Pentacene and *p*-terphenyl are aromatic molecules which adopt the triclinic and monoclinic crystal structures, respectively. Films of 1 μm thickness were investigated as a function of pentacene concentration, with morphology shown in Fig. 2a and Supplementary Fig. 1. In dilute samples, the films form micron-sized platelets comparable to those of pure *p*-terphenyl, shown in Supplementary Fig. 2. From 1% dopant concentration, small grains approximately 50 nm in diameter appear within the crystallites. Above 10%, the platelets fragment into smaller grains, exacerbated with increasing concentration, and at 50%, the crystallites are altogether more faceted. X-ray diffraction (XRD) indicates that the structure of dilute samples (<1%) is unchanged and a single sharp diffraction peak corresponding to texture along the (003) plane of *p*-terphenyl is observed (Fig. 2b). However, above 1%, broadening is observed as well as a linear peak shift towards lower scattering angles (Fig. 2c). Peak broadening indicates decreased crystalline coherence length, predominantly attributed to reduced grain size via the Scherrer equation[31], although lattice strain-induced disorder may also contribute. Due to the substitution of larger molecules within the host, a gradual increase in the lattice parameter (i.e.

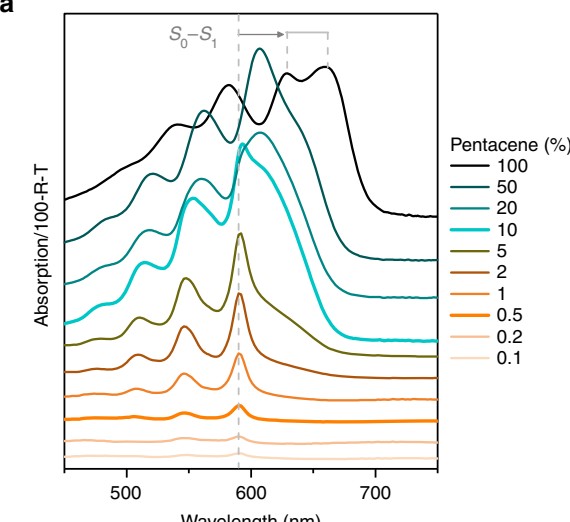

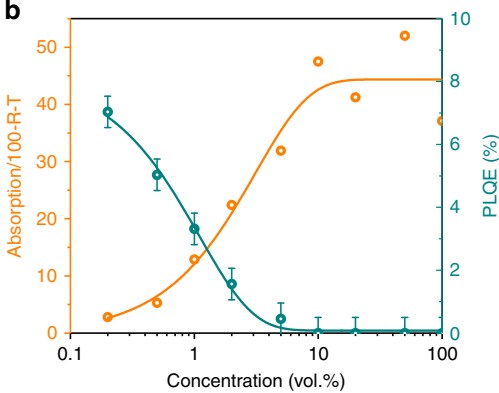

**Fig. 3** Optical spectroscopy of the pentacene doped micron films. **a** Absorption spectra (offset for clarity) as a function of pentacene concentration, with reflection and transmission subtracted from incident intensity (100-R-T), thus accounting for scattering. Below 2% pentacene, the single molecule spectra is seen with the $S_0-S_1$ transition at 590 nm, while the Davydov split peak gradually appears above this concentration. A noticeable red-shift is also observed at higher concentrations. **b** The anti-correlation with pentacene concentration of absorption at 590 nm and photoluminescence quantum efficiency (PLQE) measured at 590 nm excitation, including a 0.5% error due to instrument resolution. Trend lines are a guide to the eye. Above 10% pentacene, peak shifts affect the extracted value, whilst PLQE becomes negligible

reduced scattering angle) with increased pentacene concentration is observed, a trend that also occurs in solid solutions (i.e. a uniformly distributed minor component within the crystal lattice of its host) of inorganic materials[32–34]. Furthermore, there is no evidence for phase separation either in XRD or scanning electron microscopy (SEM), implying a homogeneous distribution of molecules within the films.

**Optical determination of aggregation.** Absorption spectra of pentacene (Fig. 3a) are highly sensitive to changes in molecular proximity and packing, and thus facilitate characterisation of aggregates (used here to describe two or more electronically interacting pentacene molecules). As pentacene concentration within the colourless $p$-terphenyl host increases, two distinct regimes are apparent. Spectra of films comprising 1% pentacene and below show an $S_0 \rightarrow S_1$ transition at 590 nm, with an associated vibronic progression at higher energies. While the total

absorption increases as expected with pentacene concentration, the symmetric peaks demonstrate that the majority of pentacene molecules are sufficiently dispersed in $p$-terphenyl to avoid intermolecular interactions (i.e. they remain unaggregated). Supporting this description is the fixed spectral location of pentacene's $S_0 \rightarrow S_1$ transition (Supplementary Note 1, Supplementary Fig. 3), indicating a lack of intermolecular electronic coupling and stable local polarizability environment. However, above 1% concentration, a lower energy shoulder attributed to Davydov splitting[35] gradually becomes more pronounced with increasing proportions of pentacene. As the concentration increases, the high-energy Davydov component shifts to lower energies along with an attendant increase in Davydov splitting (Supplementary Fig. 3).[19] Additionally, above 10% pentacene, an $S_0 \rightarrow CT$ absorption can be identified at ~560 and 610 nm[36], attributed to π-stacks with strong intermolecular interactions and nearest neighbour separations of around 3.5 Å[15,37,38]. Both the splitting magnitude and the relative intensity of the lower energy Davydov component depend on the CT interaction strength and hence intermolecular spacing since the exciton responsible for the lower energy Davydov component includes a strong (~45%) CT contribution.[7,19,39,40]

Photoluminescence quantum efficiency (PLQE), indicative of the proportion of absorbed photons that subsequently decay radiatively, provides further evidence of pentacene aggregation increasing with concentration. Figure 3b shows that up to 5% concentration, there is a clear anticorrelation between the PLQE and the absorption at the excitation wavelength of 590 nm. The initial drop in PLQE is attributed to excitons diffusing (e.g. via a Förster mechanism) to a small proportion quenching sites (i.e. aggregates) not apparent in the absorption spectra. For a concentration of 10% and above, there is negligible PLQE due to increased absorption and substantial fluorescence quenching. As we demonstrate below, aggregation-induced quenching is attributed to SF supplanting fluorescence (Supplementary Fig. 4).

**Orientation of pentacene molecules in the film.** XRD characterisation did not enable firm conclusions to be drawn about the orientation of pentacene in the host matrix. However, this could be achieved by exploiting the anisotropy of the zero-field splitting (ZFS) of the triplet state of pentacene using time-resolved EPR spectroscopy (tr-EPR)[41–43]. The ZFS is strongly orientation dependent due to its dipolar origin. Consequently, it can be used to derive structural information if the orientation of the ZFS axes are known with respect to the molecular scaffold. For pentacene, the long molecular axis is the $x$-axis, the $y$-axis is parallel to the short molecular axis and the $z$-axis is perpendicular to the aromatic plane (Fig. 1a). Further details on the theoretical basis for the experiment can be found in the Supplementary Note 2. Figure 4a shows the rotation pattern for the 0.5% film over 180°, where the angle specifies the orientation between the applied magnetic field and the normal to the substrate. The 0.5% film was used as it has the strongest ISC triplet signal, which enabled even the smallest contributions to be resolved. Simulation of the rotation pattern (coloured trace in Fig. 4a) closely resembles that obtained for a pentacene doped $p$-terphenyl crystal[26]. At $20 \pm 5°$ between the normal and the magnetic field, a single pair of peaks with a splitting of 55 mT (i.e. 1550 MHz, corresponding to the $D + 3E$ transition, where $D$ and $E$ are ZFS parameters defined in Supplementary Fig. 5) can be identified as the pure $x$-orientation. This indicates that the molecules have a contact angle of ~70° on the substrate, slightly further off the normal than the host contact angle (76°) with an experimental error of ±5°. When rotated by 90°, the $y,z$-orientations (splitting ~46 mT and ~100 mT, corresponding to the $D−3E$ and $2D$ transitions, respectively)

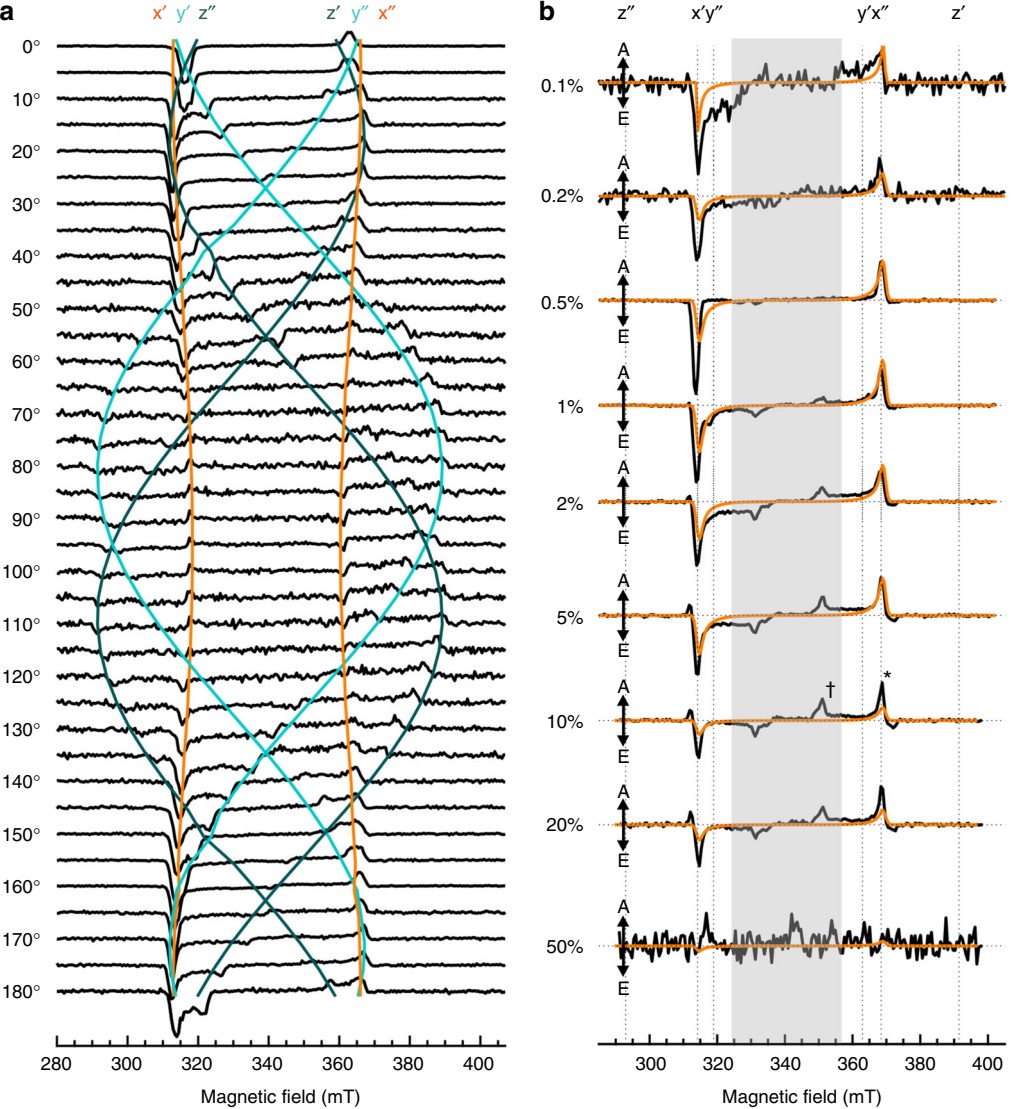

**Fig. 4** tr-EPR of ordered pentacene films as a function of orientation and concentration. **a** tr-EPR spectra of the 1 μm 0.5% pentacene doped *p*-terphenyl film at different orientations within the magnetic field taken at time 400 ns. Angles are defined as between the normal to the substrate and the magnetic field direction. The coloured traces are simulations of the rotation pattern of each molecular axes, calculated for the two inequivalent sites in the crystal lattice, where orange, cyan and teal correspond to x, y and z, respectively. **b** tr-EPR spectra acquired in the *x*-orientation as a function of pentacene concentration with the simulated triplet contribution shown in orange. The spectral slices are taken at the maximum signal, time 400 ns below 10% and at 500 ns above 10%. The inner peaks found in the shaded area, which do not belong to isolated triplets, are assigned to long-lived strongly coupled triplet states. The (*) and (†) on the 10% spectra represent the triplet and quintet absorption peaks, respectively

are resolved with no *x*-contribution. Increased in-plane disorder when the *y,z*-orientation is parallel to the field resulted in a weaker signal at those orientations. This leads to the conclusion that all pentacene molecules are aligned along the *x*-axis but that there is no preferential orientation in the *y,z* plane.

The paramagnetic excited states of pentacene were also investigated as a function of concentration, with the results for the ZFS *x*-axis (Fig. 1a) parallel to the magnetic field shown in Fig. 4b. The preferential orientation is maintained up to 20% pentacene, above which the EPR signal is quenched, suggesting that clusters of pentacene are sufficiently close to enable exciton hopping at room temperature. Outer peaks are attributed to a triplet contribution whereas the inner peaks in the shaded region, which are consistently observed above 1%, do not correspond to a texture variation but a different spin species. The appearance of the inner peaks correlates with structural and optical data, which demonstrates aggregation above this concentration, and our

subsequent analysis shows these peaks belong to quintets formed by pentacene dimers with two distinct geometries, either parallel,[5] $(TT)_{||}$, or herringbone,[5] $(TT)_{\wedge}$ (Supplementary Fig. 6).

To unequivocally identify the spin states of the species causing the observed transitions, transient nutation experiments were performed on the 10% pentacene sample, chosen for its optimum signal-to-noise ratio. These experiments allow the identification of strongly exchange-coupled triplets, where the exchange coupling $J > D$, forming quintet states with $S = 2$.[12,13] Analysis of the experiment is provided in the Supplementary Note 3. Figure 5a, b shows the nutation frequency and corresponding fast Fourier transform (FFT) of the long-lived triplet and quintet states at the high-field position, shown by (*) and (†) in Fig. 4b. The FFT shows the triplet and corresponding quintet nutation frequency with a ratio of $\sqrt{3}$, as well as an intermediate peak with ratio of $\sqrt{2}$. The former contribution is due to an $m_{S0}$ to $m_{S\pm1}$ transition, and the latter is due to an $m_{S\pm1}$ to $m_{S\pm2}$ transition.

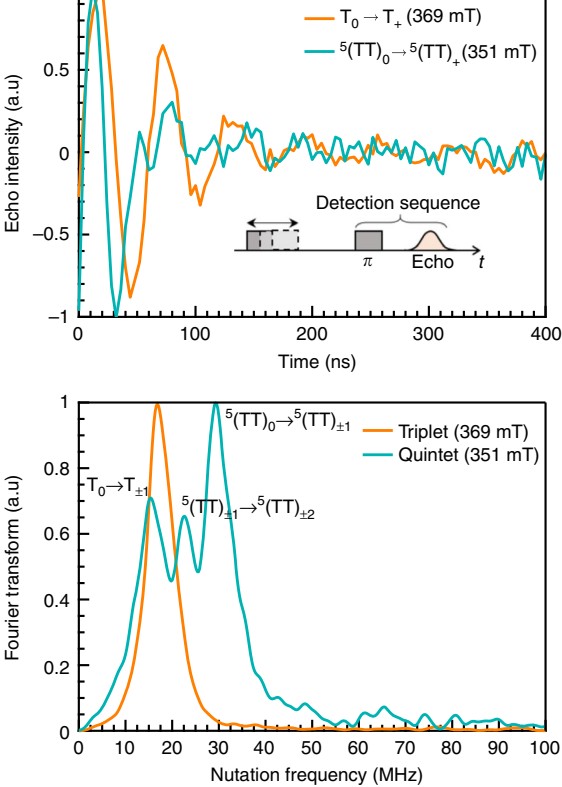

**Fig. 5** Identifying quintet states at room temperature via nutation experiments. **a** Nutation experiment for the high-field quintet and triplet transitions, with the pulse sequence shown in the inset. **b** Fourier Transform of data in **a** showing a ratio of 1.74 between the triplet and quintet nutation frequencies (16.85 and 29.3 MHz, respectively), in accord with the nominal value of $\sqrt{3} = 1.73$

Moreover, the measured nutation frequencies indicate that quintet sublevels with $m_s \neq 0$ are also populated in this system.

**Multiexciton spin states.** Close inspection of the experimental spectra for the 10% pentacene film along the $x$-orientation (20° ± 5° with respect to the applied B-field, as reported in Fig. 6) reveals four pairs of peaks, with the two inner peak separations about 1/3 of the outer peak separations. A similar scenario is also observed for the $y,z$-orientation (110° ± 5° with respect the B-field, Fig. 6). Together with the transient nutation experiment, this implies the presence of two different quintet states with distinct magnetic properties. To disentangle the individual contributions and to estimate the magnetic parameters of the excited spin states, the $x$- and $yz$-orientations were simulated simultaneously with a single set of parameters, detailed in the Supplementary Note 4. As predicted from the host crystal structure, the nearest neighbours are separated by ~4 Å edge-to-edge and can either be in a herringbone or parallel geometry. The proximity of adjacent pentacene molecules results in strong coupling, which in principle makes SF possible for both conformations. Therefore, the simulations included three contributions: (1) a pair of interacting triplets representative of the parallel conformation (teal dimer in the inset of Fig. 2b); (2) a pair of interacting triplets representative of the herringbone conformation (cyan dimer in the inset of Figs. 2b); and (3) ISC triplet stemming from isolated pentacene molecules still present at 10% doping, due to the statistical distribution of aggregate sizes in the sample as expected from the deposition technique. In the $x$-orientation, the simulation used an isotropic exchange coupling $J = 20$ GHz, with an added spin–spin

interaction component of 60 MHz for the parallel dimer configuration, as calculated in previous studies[12,44]. It should be noted the isotropic $J$-coupling stated is a minimum threshold, not a precise value.

Given that the quintet wavefunction spans two pentacene molecules, the angle between pentacene molecules in the herringbone dimer configuration was set to the average of the host and dopant lattice parameters, as an exact crystallographic structure has not yet been determined. The most precise description of the blended film to date is obtained through tr-EPR and is presented in Fig. 4. The added angular contribution to the dipolar coupling, consistent with the herringbone configuration, accounted for the phase change between the quintet and triplet peaks of the EPR spectra whereas the collinear coupling, corresponding to a parallel orientation, resulted in the same triplet and quintet phase. The outer pair of triplets and corresponding quintet could, therefore, be attributed to the herringbone dimer and the inner pairs to the parallel dimer. Dimerisation has been shown to alter the ZFS parameters with respect to the constituent monomers although no clear link to morphology has been made[14,45]. Here, we conclude that the parallel dimer has a ZFS unchanged with respect to the monomer, whereas the ZFS increases in a herringbone conformation.

In the $yz$-orientation, in-plane disorder added complexity to the model. The simulation reported in Fig. 6 used a 90° rotation of the $x$-orientation calculation, keeping all coupling parameters the same and correctly attributing experimental peak positions to $y,z$ molecular orientations. However, the two innermost peaks correspond to an $x$-orientation quintet, following the trend that increased concentration leads to increased disorder. An $S = 2$ model for a polycrystalline sample was also calculated (Supplementary Fig. 7), which includes an ordering parameter that accounts for the increased texture. With this method, the full spectrum was simulated but no precise $J$ value was computed. Table 1 reports a comparison with literature data on analogous systems. The magnetic parameters derived here for ordered films compare satisfactorily with those obtained from amorphous films and frozen solutions.

With the unprecedented detection of specific dimer configurations, it has been possible to assign distinct kinetic parameters to the various spin states. We therefore report two different behaviours according to whether dimers are in a parallel or herringbone geometry, with details of the kinetic model provided in the Supplementary Note 5. Figure 7a depicts a 2D mapping of the tr-EPR spectrum of the 10% pentacene film with the applied magnetic field parallel to the $x$-axis and a corresponding scheme in Fig. 7b, detailing the time evolution of triplet dynamics as derived from our data. The corresponding spin populations are shown in Supplementary Fig. 8. The decay of the parallel quintet lifetime ($^5(TT)_\parallel$ the innermost peaks at 334 and 349 mT) coincides with a rise in the corresponding triplet signal (at 315 and 369 mT). This implies a short-lived high-spin state that efficiently separates into free triplets, which in turn are long lived (>3 μs). The modelled time traces are shown in Fig. 7c, with the normalised data shown to present all positive traces. In the parallel geometry, the rate of dissociation, $8.0 \times 10^6$ s$^{-1}$, is an order of magnitude faster than the T + T → $^5(TT)_\parallel$ back transfer reaction. It should be noted that in the parallel configuration, the dissociated triplet contribution overlaps with the ISC triplet signal, which has been accounted for in the model and appears faster than in non-aggregated samples.[46] Conversely, the herringbone quintets ($^5(TT)_\wedge$ at 331 and 351 mT) coexist with the weak, outermost triplet signal (at 312 and 372 mT), due to inefficient dissociation. Indeed, the rate of dissociation, $8.0 \times 10^6$ s$^{-1}$, matches the rate of the T + T → $^5(TT)$ back reaction, $1.6 \times 10^7$ s$^{-1}$, with the factor two stemming from two

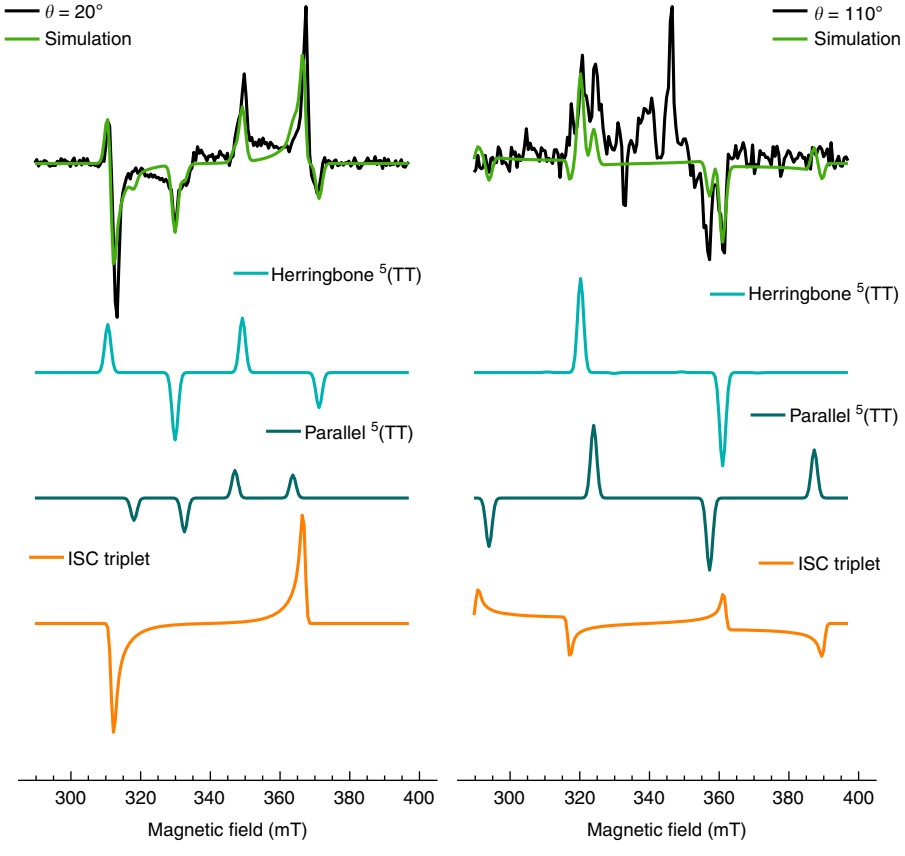

**Fig. 6** Evidence of multiple spin species resulting from singlet fission at room temperature. The experimental spectra (black) can be fitted (green) with a superposition of simulated peaks for the herringbone (cyan) and parallel (teal) dimer arrangement of strongly coupled triplets as well as the intersystem crossing triplet (orange). For the 110° orientation, the additional peaks that are not simulated are due to in-plane disorder, as verified using an $S = 2$ fit with an ordering parameter shown in Supplementary Fig. 7. Spectral slices were taken at the maximum signal, time = 500 ns

**Table 1 Comparison of the fitted parameters derived in this work with previously reported data on analogue systems**

| Sample | Form | Temperature (K) | $J_{iso}$ (GHz) | X (MHz) | \|D\| (MHz) | \|E\| (MHz) | Ref. |
|---|---|---|---|---|---|---|---|
| Pentacene | Oriented film | | | | | | This work |
| | (herringbone) | 298 | >20 | - (a) | 1600 | 80 | |
| | (parallel) | 298 | >20 | 60 | 1400 | 50 | |
| TIPS-tetracene | Amorphous film | < 75 | 1.02x10³ (b) | - | - | - | 13 |
| BP2 Dimer | Frozen Solution | 80 | 29.2 | 10 | 1078 | 13 | 12 |
| BP3 Dimer | Frozen Solution | 20 < T < 80 | 19.9 | 39 | 1138 | 19 | 12 |
| NC-m Dimer | Frozen Solution | 5 & 105 | - | - | 1200 | - | 46 |
| NC-p Dimer | Frozen Solution | 5 & 105 | - | - | 1200 | - | 47 |
| SPi Dimer | Frozen Solution | 80 | >20 | - | 1100 | 20 | 48 |
| BCO Dimer | Frozen Solution | 80 | >20 | - | 1100 | 20 | 48 |

(a) Herringbone dimer is treated as isotropic and parallel dimer has large anisotropy.
(b) Estimated upper limit

triplets formed for every quintet. This could result from a thermally activated equilibrium between the quintet and triplet states. Due to the inefficient dissociation, these would result in trap-states within the functional layer of an optoelectronic device.

## Discussion

SF leads to the generation of excited triplet and quintet states on pairs of interacting chromophores. Only recently has quintet formation been observed by EPR spectroscopy at cryogenic temperatures from pentacene dimers in solution and TIPS-tetracene cast as amorphous films. A major ambiguity in the understanding of SF stems from the poor experimental correlation between molecular geometry and spin coupling. We

exploited our growth method to bridge the gap between covalent dimers in solution and monomers in films but crucially at room temperature and with highly controlled molecular orientation. The use of ordered structure and a progressive increase in pentacene concentration allowed the unique observation and characterisation of two strongly coupled quintet states. Moreover, dilution inhibits excitation/spin diffusion, significantly enhancing the excitation lifetime. An optimal dilution would provide a compromise between SF efficiency and competing pathways. We have provided direct evidence for two distinct spin coupled states that can be assigned to pairs of pentacenes by their relative orientations. This unambiguously proves that: (a) quintet states at room temperature in pentacene are long lived and their presence

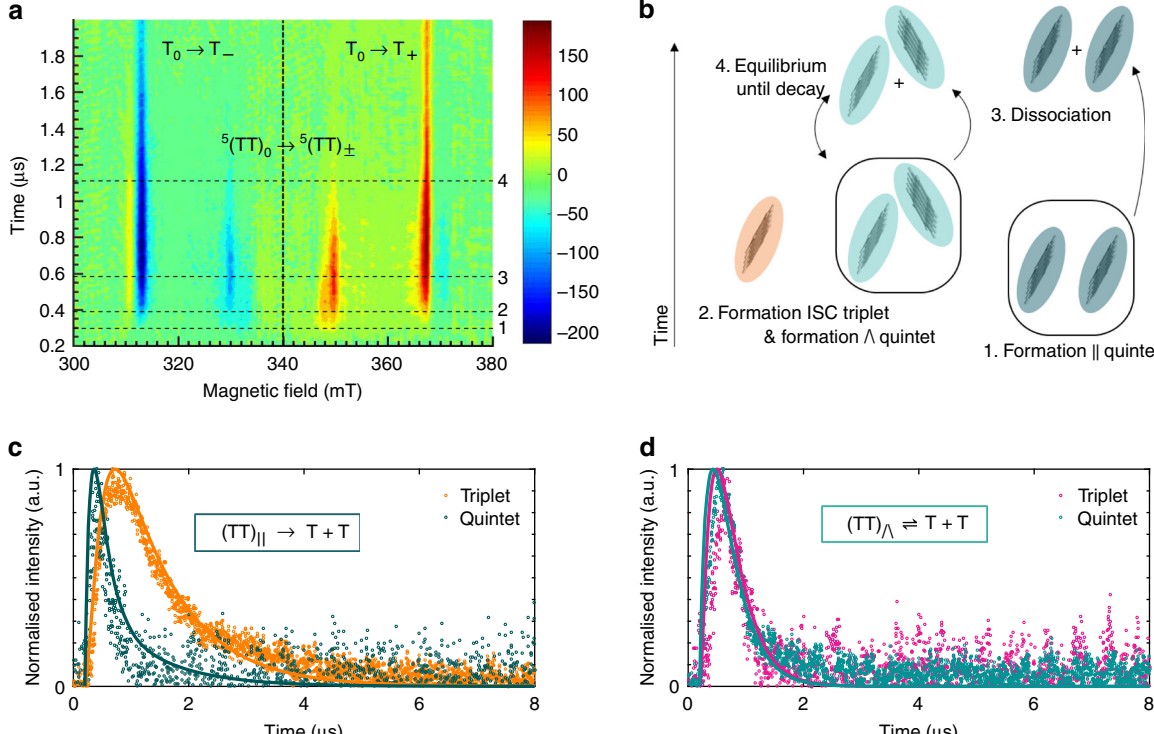

**Fig. 7** Dynamics of the triplet and quintet states. **a** Tr-EPR of the 10% pentacene in *p*-terphenyl 1 μm film. The innermost peaks from the centre-field belong to strongly coupled triplets from parallel dimers, $^5(TT)_{||}$, with the next pair of peaks originating from strongly coupled triplets with a herringbone geometry, $^5(TT)_{//}$. The penultimate and strongest pair of peaks belong to dissociated parallel triplets and isolated ISC crossing triplets, (T). The outermost peaks are the dissociated triplets from the herringbone quintet. Timescales for the formation and dissociation of spin species are labelled 1–4 on the right-hand side. **b** Schematic of the triplet and quintet kinetics: (i) formation of $^5(TT)_{||}$, (ii) formation of ISC (T) and of the $^5(TT)_{\wedge}$, (iii) dissociation of the quintets into triplets, (iv) equilibrium state between $(TT)_{\wedge} \rightleftharpoons T + T$, forming a trap-state, whilst $(TT)_{||} \rightarrow T+T$ forms long-lived triplets. **c, d** Show the experimental (circles) and modelled (line) time traces of the parallel and herringbone geometries, respectively. Experimental traces in emission have been inverted for clarity

should be included in the analysis of ultrafast optical experiments; (b) the geometry of the pentacene dimers directly affects the properties and dynamics of the corresponding triplet and quintet high-spin states; and (c) specific geometries are likely to promote the efficiency of SF and to extend the lifetime of the triplet excitons. We have demonstrated that the parallel configuration of dimers undergoes efficient dissociation, $(TT)_{||} \rightarrow T+T$, and is therefore most favourable to generate charges in photovoltaic devices. In contrast, the herringbone configuration results in a trap-state that would hinder triplet harvesting in a device, $(TT)_{\wedge} \rightleftharpoons T + T$, although the strong coupling on microsecond timescale could provide opportunities for the exploitation of spin interactions in memory applications. In light of the results presented here, selecting and controlling specific geometries of correlated triplet pairs can be used to design more efficient triplet harvesting layers; specifically altering the structure into a brick-stack arrangement would lead to a reduction in trap-states and improve optoelectronic device efficiency.

## Methods

**Thin film fabrication**. Commercially obtained pentacene purified by sublimation (TCI UK Ltd.) and *p*-terphenyl (99+% Alfa Aesar) further zoned refined were deposited by organic molecular beam deposition using a Kurt J. Lesker SPECTROS 100 system with a base pressure of $10^{-7}$ mbar. Quartz crystal microbalance sensors were used to precisely control the deposition rates and therefore the concentration of the thin films. The concentrations are expressed as the ratio by volume of pentacene relative to *p*-terphenyl with the precise values to within 0.1% as follows: 0.1%, 0.2%, 0.5%, 1.1%, 2.0%, 4.8%, 9.0%, 16.8%, 50.2%, 100%. For simplicity, these are referred in the main text as: 0.1, 0.2, 0.5, 1, 2, 5, 10, 20, 50 and 100%.

**Scanning electron microscopy**. The morphology was investigated using a high-resolution LEO Gemini 1525 field-emission gun scanning electron microscope (FEGSEM). To obtain the cross-sections, thin films deposited on Si(100) were cleaved to obtain a sharp surface.

**X-ray diffraction**. For the structural measurements, a PANalytical X'Pert Pro multi-purpose diffractometer was used with a Cu $K_\alpha$ X-ray source. The grain size was calculated using the Scherrer equation, with the difference in broadening between a Gaussian and Lorentzian fit used as the uncertainty in calculation[47]. Large grains have a much larger error as these are at the instrument limit.

**Absorption and PLQE**. Absorption spectra were determined using a Shimadzu UV-2600 spectrometer fitted with an integrating sphere, enabling scattered light to be collected, thus removing this loss contribution from the absorption spectra. Furthermore, the total reflection (i.e. diffuse and specular) was measured, using a negligibly absorbing fused silica substrate as a reference, and subtracted along with transmission from 1 (i.e. total incident intensity), leaving only absorption.

Photoluminesence spectra and the corresponding PLQE values were determined by placing the thin film sample within a 15-cm diameter integrating sphere (Labsphere, internally coated with Spectraflect), before excitation using a monochromated supercontinuum Fianium light source at 590 nm. Both excitation and emission were transferred using a 100-μm diameter optical fibre to an Andor SR-163 spectrometer and recorded with an Andor i-Dus CCD. Integrating sphere reflectivity and CCD response was corrected for using a calibrated halogen light source (HL-2000-CAL, Ocean Optics). PLQE spectra was determined using the de Mello method[48], by comparing the integrals of excitation (attenuated by absorption) and emission peaks for direct and indirect excitation along with the excitation peak with an empty sphere. PLQE error bars are estimated at 0.5%.

**EPR spectroscopy**. All EPR measurements were acquired using a Bruker E580 pulsed EPR spectrometer operating at X-band frequencies (9–10 GHz/0.3 T), operating either in continuous wave time-resolved (cw) or pulsed mode, equipped with an Oxford Instruments CF935O flow cryostat used to support a Bruker ER4118-X MD5 resonator. The microwave frequencies used in cw experiments for

concentrations between 0.1 and 20% pentacene in *p*-terphenyl were (in order, in GHz): 9.542, 9.544, 9.580, 9.578, 9.617, 9.620, 9.543, 9.538 and 9.541. The microwave frequency used in the pulsed experiment for the 10% sample was 9.541 GHz.

A Surelite broadband OPO system within the operating range 410–680 nm, pumped by a Surelite I-20 Q-switched Nd:YAG laser with 2nd and 3rd harmonic generators (20 Hz, pulse length: 5 ns), was used to achieve a pulsed laser excitation at 580 nm, the maximum absorption for pentacene, with the energy at the sample approximately 5 mJ per pulse.

All spectra were recorded at room temperature in air atmosphere.

**tr-EPR spectroscopy**. tr-EPR spectra were recorded in direct detection mode without magnetic field modulation. Hence, they show characteristic enhanced absorptive (A) and emissive (E) features. The rotation pattern, used to derive the orientation of pentacene molecules within the sample, was recorded by means of a laboratory-built goniometer in step of $10 \pm 2°$. The sample was mounted on a bespoke quartz support designed to keep the film slide upright with the long axis parallel to the axis of the resonator. TR-EPR spectra were simulated using the function *pepper* present in the EasySpin[49] toolbox running on MATLAB™. Oriented spectra were calculated as a superposition of spectra of the two inequivalent sites, related by crystallographic (P $2_1$/a), taking as starting values those reported for bulk pentacene:*p*-terphenyl crystal at room-temperature[50,51].

The simulation parameters include the ZFS tensor principal values and the orientation of ZFS for one site in the unit cell, the orientation of the crystal in the magnetic field, and a line width parameter that accounts for the unresolved hyperfine interactions. Given that the ZFS tensor is traceless, its principal values $D_x$, $D_y$ and $D_z$ can also be expressed by two ZFS parameters $D$ and $E$. These are related to the principal values according to the relations: $D_x = 1/3D + E$, $D_y = 1/3D - E$ and $D_z = -2/3D$. ZFS parameters and sublevel populations were taken from literature values, with $D = 1400$ MHz, $E = 50$ MHz[52] and $P_x:P_y:P_z = 0.76:0.16:0.08$[53]. An isotropic *g*-value equal to the free electron *g* value ($g_x = g_y = g_z = 2.0023$) was used in all simulations.

**Pulsed EPR spectroscopy**. Echo-detected field sweep spectra (not shown) were recorded using the pulse sequence $\pi/2$-200-$\pi$-200-echo with a $\pi/2$ pulse of 16 ns. Transient nutation spectra were recorded at fixed magnetic fields with the shortest delay after flash available ~100 ns and pulse sequence $P_{nutation}$-400-$\pi$-400-echo, with $P_{nutation}$ starting at 4 ns and incrementing by 4 ns for 101 points. The resulting echo was integrated in quadrature detection as a function of nutation pulse length, $P_{nutation}$. The time-domain data were baseline corrected with a second-order polynomial function, tempered with and Hamming window function, zero-filled and Fourier transformed to establish the nutation frequency of a given transition. The absolute part is reported in the figure.

## Data availability

The data and code that support the findings of this study are available from the corresponding authors upon request.

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

## Acknowledgements

D.L.P. acknowledges a studentship from the UK Engineering and Physical Sciences Research Council (EPSRC) Centre for Doctoral Training for the Advanced Characterisation of Materials (EP/L015277/1) and a research placement supported by the joint research program of Molecular Photoscience Research Center, Kobe University, under the supervision of Y.K. R.M. thanks the EPSRC Fellowship (EP/K037390/1) for funding. M.D. thanks the UK EPSRC Plastic Electronics Doctoral Training Centre (EP/G037515/1) and the Marie Sklodowska-Curie Actions Innovative Training Network "H2020-MSCA-ITN-2014 INFORM—675867" for funding.

## Author contributions

R.M. purified the pentacene and *p*-terphenyl. The films were fabricated and characterised by D.L.P. Optical spectroscopy was carried out by M.D. and D.L.P. under the supervision of P.N.S. M.D. performed the photoluminescence calculations. D.L.P., E.S. and C.W.M.K. carried out the EPR spectroscopy measurements. Y.K. and H.N. provided insights into the formalism of the SF mechanism. D.L.P. and E.S. interpreted the experimental results. D.L.P. prepared the figures. D.L.P. and E.S. wrote the manuscript with critical feedback from all authors. The study was conceived by S.H. and C.W.M.K. and supervised by E.S., S.H. and C.W.M.K.

## Additional information

**Competing interests:** The authors declare no competing interests.

