## [peer review file · Nature Communications]

Reviewer #1 (Remarks to the Author):

This paper is an important contribution to the growing EPR literature on singlet fission. This paper is particularly clearly written, and the analysis of the data follows established EPR protocols. It is interesting that the two different pentacene dimer morphologies show different rates of T + T formation from the quintet TT state. While I don't think that pentacene has any long-term device prospects (this aspect is perhaps a little too strongly emphasized in the paper), the fact that the authors have demonstrated that different dimer morphologies can result in different propensities of the TT pair to separate is a very important demonstration for the field.

I worked my way through the analysis in this paper twice and everything looks sound to me. I usually have many comments and concerns when reviewing singlet fission manuscripts, but this one is a gem of clarity. It should be published.

Reviewer #2 (Remarks to the Author):

Identifying triplet pathways in organic semiconductors is certainly a timely objective. Here, this is addressed by looking at host-guest systems with increasing concentration of pentacene (SF active) molecules in an inert (large bandgap) terphenyl host. In that respect, the approach is very much similar to that reported by Broch et al in Ref. 19. The use of EPR to selectively assess triplet and quintet states has also been demonstrated, namely in the work by Weiss et al in Ref. 13. Thus, one could argue about the novelty of this work. But that would be fine if the authors would have made a clear and comprehensive case that their work goes beyond the current state of the art, and I fear this is not the case. First because the paper is written in an incomprehensible manner (at least to non-EPR experts). The discussion is often very confusing, i.e. the whole section on optical determination of aggregation is difficult to read and highly speculative. Namely, the appearance of two CT states in more concentrated samples and the evolution of the Davydov Splitting with pentacene concentration. Second, because I doubt the experimental data at hand allow distinguishing dimers from larger aggregates, and even more so herringbone versus parallel dimers. Third, because a lot of important material has been moved out from the main text to the supporting information. Overall, I am convinced there are interesting findings in this work but, for sure, it lacks novelty and scholarship presentation for publication in Nature Communications. For the work to be really useful to the community, I would urge the authors to prepare a long version of their manuscript where most of the SI material is moved in the main text and an extensive description of

the EPR technique is provided and to submit their revised ms to a more specialized journal with focus on physical chemistry.

Reviewer #3 (Remarks to the Author):

The authors have provided an electron spin resonance (ESR) study of pentacene aggregates within a terphenyl matrix. They have shown that two pentacene-pentacene geometries exist: herringbone and parallel. The work is an impressive combination of complex sample preparation & optical characterisation and spin dynamics probed using tr-ESR.

The work is novel and provides insights about the recent plethora of tr-ESR studies of singlet exciton fission. In particular, the angular dependence of the data and the agreement with simulations are commendable.

I do, however, have several issues with the paper in its current form and recommend the following areas of improvement. Overall I think that this work is of interest to the field, but I would like clarity about some of the conclusions.

1. Figure 1: I have several issues with Figure 1. Why is only the quintet manifold drawn for the (TT) state? is the triplet (TT) state supposed to be $3T + 3T$, such that they are uncoupled triplets. Where is the $1(TT)$ state, and, since fission is spin-allowed, shouldn't it be to the $1(TT)$ state? Also why is fission drawn to proceed to the $5(TT)+2$ state. A high-field basis is used without any explanation. The diagram also lacks the direct decay of the (TT) state to the $S0+S0$ state.

2. Page 5, paragraph 1, final sentence: I think I understand what the authors are trying to say, but it is unclear. Perhaps they could reword it to state why they chose to present data with 0.5% pentacene in figure 4, and why they used 10% in Figure 5?

3. I am not convinced by their kinetic analysis. Ideally we would have rates for the scheme in Figure 5b rather than fits to decays in Table S1. For instance, how quickly is $5(TT)$ feeding $T+T$? Further, how can they prove that $5(TT)$ is equilibrating with or dissociating into $T+T$ rather than $5(TT) \rightarrow S0+T$? I do not think a proof is necessary for the overall aim of the paper, but it would be good to mention the possibility in Fig 5b.

4. Further to the above, at 10% doping there is complete suppression of the PL (Fig 3a). Yet in Figure 5, they claim to observe a triplet formed by intersystem crossing. If PL is quenched (lifetime ~ 10 ns), then there is a process which is much faster (singlet fission) which precludes the existence of intersystem crossing. Therefore there are two possibilities: (i) there are a subset of lone pentacene molecules or (ii) this signature belongs to a different species. Given the time profile I would suggest the latter is more likely than the former, but without access to the data I cannot be sure. I would however, suggest the authors consider the population of $5(TT)^{+/-1}$ since they clearly see it in the nutation data.

5. Please use a more appropriate colour scale (with more contours) in Fig 5a. It is very difficult to see the low intensity data.

6. Please include the exact microwave frequency of each ESR experiment in the figure captions or the methods section.

7. Can the authors comment on the stability of the samples? Cofacial pentacene is known to dimerize when photoexcited, and this gives rise to naphthalenic absorption bands. If these are observed in exposed samples it could provide insight into degradation mechanisms and importantly the geometry of parallel dimers.

8. The spectral slices in Figure S9 do not have a time label.

9. In Table 1 the bipentacenes are conventionally labelled BP2 and BP3. I would also point the authors toward other pentacene dimer tr-ESR papers to fill out that table: Nature Communications volume 8, Article number: 15171 (2017); J. Am. Chem. Soc., 2017, 139 (36), pp 12488–12494 & Chem Volume 4, Issue 5, p1092–1111.

Reviewers' comments are reported in blue

Our response is reported in black

Reviewer #1 (Remarks to the Author):

This paper is an important contribution to the growing EPR literature on singlet fission. This paper is particularly clearly written, and the analysis of the data follows established EPR protocols. It is interesting that the two different pentacene dimer morphologies show different rates of T + T formation from the quintet TT state. While I don't think that pentacene has any long-term device prospects (this aspect is perhaps a little too strongly emphasized in the paper), the fact that the authors have demonstrated that different dimer morphologies can result in different propensities of the TT pair to separate is a very important demonstration for the field.

I worked my way through the analysis in this paper twice and everything looks sound to me. I usually have many comments and concerns when reviewing singlet fission manuscripts, but this one is a gem of clarity. It should be published.

We would like to thank Reviewer #1 for their comments and appreciate that our efforts to present a clear explanation were recognised.

We agree that pentacene as such is perhaps not the most suitable candidate for solar cell devices, but it is a well-established benchmark in the field and hence was chosen for this study. The points we wanted to make are (1) that the intermolecular arrangement affects electronic coupling even for triplet states and (2) that quintet states are not just a spectroscopic curiosity detectable at cryogenic temperature but that they contribute to room temperature properties. This should be carefully taken into account when analysing optical data or attributing bands to triplet excitons. As such, pentacene is still a useful reference to develop hypothetical device concepts.

On the basis of the comments received from Reviewers #2 and #3 we have added further details and comments that we hope will make our conclusions even more sound.

Reviewer #2 (Remarks to the Author):

Identifying triplet pathways in organic semiconductors is certainly a timely objective. Here, this is addressed by looking at host-guest systems with increasing concentration of pentacene (SF active) molecules in an inert (large bandgap) terphenyl host. In that respect, the approach is very much similar to that reported by Broch et al in Ref. 19. The use of EPR to selectively assess triplet and quintet states has also been demonstrated, namely in the work by Weiss et al in Ref. 13. Thus, one could argue about the novelty of this work. But that would be fine if the authors would have made a clear and comprehensive case that their work goes beyond the current state of the art, and I fear this is not the case.

We thank Reviewer #2 for offering an alternative standpoint to our work. We have taken on board their general comments and we hope they will find the resubmitted manuscript of improved clarity. We have addressed all the specific points below.

First because the paper is written in an incomprehensible manner (at least to non-EPR experts).

To address Reviewer #2's concern, we have added a comment on the relationship between zero-field splitting measured in EPR spectroscopy:

“The zfs is strongly orientation dependent due to its dipolar origin. Consequently, it can be used to derive structural information if the orientation of the zfs axes are known with respect to the molecular scaffold. For pentacene, the long molecular axis is the X-axis, the Y-axis is parallel to the C-H bond direction and the Z-axis is perpendicular to the aromatic plane (Fig. 1a). Further details on theoretical basis for the experiment can be found in the supplementary information (Figure S6).”

In addition, an introduction to EPR spectroscopy of triplet state has now been included in the SI (Figure S6). This covers the basic interpretation of EPR spectra of photoexcited triplet and quintet states as well as an explanation as to how the zero-field splitting parameters provide irrefutable evidence of precise molecular orientations. A more general and in-depth introduction to EPR, which is outside the scope of a research paper, can be found in a review recently published by one of the authors (Roessler M.M., Salvadori E., Principles and Application of EPR Spectroscopy in the Chemical Sciences, Chem. Soc. Rev., 47, 2534-2553, 2018) while a review from Weber provides an excellent overview on transient EPR (Weber S., Transient EPR, eMagRes, 6, 255-270, 2017). To help readers who are unfamiliar with the technique and encourage a wider range of research communities to follow our approach these have also been included in the references list (ref. 42 and 43 respectively).

The discussion is often very confusing, i.e. the whole section on optical determination of aggregation is difficult to read and highly speculative. Namely, the appearance of two CT states in more concentrated samples and the evolution of the Davydov Splitting with pentacene concentration.

The section on optical aggregation has been rephrased in an attempt to clarify our discussion and Fig 3a and S3 have been modified. To highlight differences in the dilute versus concentrated regimes of our samples, a plot of the position of the S0 → S1 transition as a function of concentration has also been included in Figure S3.

We would also like to emphasise that the appearance of CT states with aggregation is well established and previously pointed out by Rao *et al.* (ref. 36), which references back to electroabsorption modulated experiments (Bässler, Chem. Phys., 61, 125, 1981). With regards to the Davydov splitting, the values were extracted following the same procedure as Broch *et al.* (ref. 19) and the methodology is clearly detailed in the supplementary information (Fig. S3a, b). The results reported in our manuscript follow the same trend observed by Broch *et al.*, despite the differences in lattice parameters of the host. A more detailed discussion can now be found accompanying the revised figure in the SI.

In light of these adjustments, we hope to convince reviewer #2 that our analysis is not speculative but rather is well supported by a sound analysis and previous literature (with added references 38-40).

Second, because I doubt the experimental data at hands allow distinguishing dimers from larger aggregates, and even more so herringbone versus parallel dimers.

We agree with Reviewer #2's comments, but we note that we never claimed to be able to distinguish between dimers from larger aggregates, which are present considering the statistical

doping yielded by the deposition method. We also note that SF requires at least two interacting chromophores, but it is neither suppressed nor adversely affected by larger aggregates.

On the other hand, we are confident in maintaining that EPR spectroscopy can distinguish between herringbone versus parallel configurations. Indeed, the nature and strength of magnetic anisotropies detectable through EPR have been used with much success in a wide range of fields to derive structural information. The technique is powerful enough to pinpoint and differentiate paramagnetic centres even in amorphous samples. Examples include:

Point defects in amorphous silicon and silicon carbide: Cantin J.L. *et al.*, Identification of the Carbon Dangling Bond Center at the 4H-SiC/SiO₂ Interface by an EPR Study in Oxidized Porous SiC, *Phys. Rev. Lett.* 92, 015502, 2004.

Colour defects in diamond and silicon carbide: Ligner Th. *et al.*, Structure of the silicon vacancy in 6H-SiC after annealing identified as the carbon vacancy-carbon antisite pair, *Phys. Rev. B*, 64, 245212, 2001.

Photosynthetic reaction centres (PSII): Kammel M. *et al.*, Photosystem II single crystals studied by transient EPR: the light-induced triplet state, *BBA – Bioenergetics*, 1605 (1-3), 47-54, 2003.

Furthermore, it is undeniable that the experimental data (Fig 6 and 7a) show the presence of two distinct species. We also note that Reviewer #3 appreciates the relevance and the insight provided by the angular dependence and the simulation of our experimental data.

Third, because a lot of important material has been moved out from the main text to the supporting information. Overall, I am convinced there are interesting findings in this work but, for sure, it lacks novelty and scholarship presentation for publication in *Nature Communications*. For the work to be really useful to the community, I would urge the authors to prepare a long version of their manuscript where most of the SI material is moved in the main text and an extensive description of the EPR technique is provided and to submit their revised ms to a more specialized journal with focus on physical chemistry.

We disagree with Reviewer#2 on these comments. This paper represents the first observation of quintet states formed upon SF under ambient conditions. Moreover, it demonstrates that the intermolecular geometry also affects the electronic coupling in the triplet manifold. Although this is in principle expected, the number of experimental reports or theoretical treatments in the literature is sparse and does not provide the type of clear correlation between dimer geometry and triplet pathways that could inform for example crystal engineering or new device concepts. Thus, we believe this work will spark additional research effort both theoretical and experimental and is definitely suited to the interdisciplinary readership of *Nature Communications*.

We also feel the importance of this work is reinforced by Reviewer #1 and #3's recognition of the novelty and clarity of our work.

Following clarifications as addressed above, we believe that the paper provides an appropriate level of detail to engage a broad readership and to provide a full picture of our conclusions, spanning materials fabrication, optical characterisation and simulation including a kinetic model.

Reviewer #3 (Remarks to the Author):

The authors have provided an electron spin resonance (ESR) study of pentacene aggregates within a terphenyl matrix. They have shown that two pentacene-pentacene geometries exist: herringbone and parallel. The work is an impressive combination of complex sample preparation & optical characterisation and spin dynamics probed using tr-ESR.

The work is novel and provides insights about the recent plethora of tr-ESR studies of singlet exciton fission. In particular, the angular dependence of the data and the agreement with simulations are commendable.

I do, however, have several issues with the paper in its current form and recommend the following areas of improvement. Overall I think that this work is of interest to the field, but I would like clarity about some of the conclusions.

We would like to thank Reviewer #3 for appreciating our efforts, acknowledging the novelty of this work and providing a list of fair comments and useful suggestions. We have taken them on board and provide details below:

1. Figure 1: I have several issues with Figure 1. Why is only the quintet manifold drawn for the (TT) state? is the triplet (TT) state supposed to be $3T + 3T$, such that they are uncoupled triplets. Where is the $1(TT)$ state, and, since fission is spin-allowed, shouldn't it be to the $1(TT)$ state? Also why is fission drawn to proceed to the $5(TT)+2$ state. A high-field basis is used without any explanation. The diagram also lacks the direct decay of the (TT) state to the $S0+S0$ state.

The (TT) state was indeed supposed to be the coupled $1(TT)$ state but was mislabelled. Therefore, not only the quintet manifold is shown and the $1(TT)$ state is included. Our intention was to show the fission going from the singlet state to the $1(TT)$ state, which can then undergo mixing forming the quintets ending in the dissociation. We have amended the diagram to make this clearer. We observe the quintet states via nutation experiments, however we realised the high-field basis was unjustified. We have removed reference to this to avoid over-complication. We have also added the missing decay.

2. Page 5, paragraph 1, final sentence: I think I understand what the authors are trying to say, but it is unclear. Perhaps they could reword it to state why they chose to present data with 0.5% pentacene in figure 4, and why they used 10% in Figure 5?

This section has been rewritten to address the comments from Reviewer #2. We hope this sentence has now been made clear in the adjusted context. In addition, we have specified our choice of samples, namely 0.5% in Fig. 4 and 10% in Fig. 5 in the relevant sections as follows:

“The 0.5% film was used as it has the strongest triplet signal, which enabled even the smallest contributions to be resolved.”

“To identify the spin species unequivocally, transient nutation experiments were performed on the 10% pentacene sample, chosen for its optimum signal-to-noise ratio on the quintet peak.”

3. I am not convinced by their kinetic analysis. Ideally we would have rates for the scheme in Figure 5b rather than fits to decays in Table S1. For instance, how quickly is $5(TT)$ feeding $T+T$? Further, how can they prove that $5(TT)$ is equilibrating with or dissociating into $T+T$ rather than $5(TT) \rightarrow S_0+T$? I do not think a proof is necessary for the overall aim of the paper, but it would be good to mention the possibility in Fig 5b.

The time traces and corresponding decay fits have been removed from the SI and replaced with a kinetic model derived from coupled rate equations in the main text. The details of the analysis have been added to the SI with a table of rates (table S2) and a schematic of the states (Figure S10). We would like to highlight that this model includes the dissociation $5(TT) \rightarrow T+T$, recombination $5(TT) \rightarrow S_0S_0$, back reaction $T+T \rightarrow 5(TT)$ and triplet decay $T \rightarrow S_0$. We have also amended the Figures in the main text, dividing it in two with the nutation experiments first, Figure 5, and then the kinetics, Figure 7. This latter figure now includes the time evolutions and corresponding models. From this we hope to convince Reviewer #3 that indeed the herringbone geometry leads to a slower dissociation with the back reaction of a matched rate, whereas the parallel geometry dissociates efficiently with little back reaction. The contribution of the ISC triplet in the parallel configuration was confirmed but is not present in the herringbone geometry, as expected from the tr-EPR data. We feel this supports our claims that the herringbone dimer acts as a trap that will hinder optoelectronic applications (yet offer some potential uses in alternative spin-based device concepts) whereas the parallel geometry would be a preferred geometry for fission.

4. Further to the above, at 10% doping there is complete suppression of the PL (Fig 3a). Yet in Figure 5, they claim to observe a triplet formed by intersystem crossing. If PL is quenched (lifetime ~ 10 ns), then there is a process which is much faster (singlet fission) which precludes the existence of intersystem crossing. Therefore, there are two possibilities: (i) there are a subset of lone pentacene molecules or (ii) this signature belongs to a different species. Given the time profile I would suggest the latter is more likely than the former, but without access to the data I cannot be sure. I would however, suggest the authors consider the population of $5(TT)+/-1$ since they clearly see it in the nutation data.

We would like to thank Reviewer #3 for identifying this apparent contradiction. Figure S4 shows that the 10% pentacene sample does fluoresce slightly under excitation, although given its greater absorption this results in a negligible PLQE. In addition to the statistical distribution of aggregate sizes due to the deposition technique, this supports the former suggestion, which is the presence of a subset of isolated pentacenes. Furthermore, it has been reported that pentacene dimers in a *p*-terphenyl host can undergo ISC orders of magnitudes faster than isolated pentacene, which also supports the presence of these triplets. To address this, we have added a comment and ref. 48 to the main text.

We would also like to thank Reviewer #3 for suggesting the alternative interpretation of other spin species, namely the $5(TT)+/-1$. Indeed, during our theoretical simulations we initially investigated the possibility of this contribution, because this state appears to be populated in nutation experiments. However, this did not agree with our experimental traces and consequently we turned to the theory presented in this paper.

5. Please use a more appropriate colour scale (with more contours) in Fig 5a. It is very difficult to see the low intensity data.

We have changed the colour scheme and added more contours to make the low intensity data more distinguishable. The tr EPR data is now part of Figure 7.

6. Please include the exact microwave frequency of each ESR experiment in the figure captions or the methods section.

The microwave frequencies for each experiment have been added to the methods section with corresponding concentration.

7. Can the authors comment on the stability of the samples? Cofacial pentacene is known to dimerize when photoexcited, and this gives rise to naphthalenic absorption bands. If these are observed in exposed samples it could provide insight into degradation mechanisms and importantly the geometry of parallel dimers.

The samples are extremely stable as the p-terphenyl passivates the pentacene. This has been demonstrated in several maser experiments (Salvadori E, *et al.*, *Sci Rep*, 7, 41836, 2017; Breeze J.D., *et al.*, *Nat Comm*, 6, 6215, 2015; Oxborrow M., *et al.*, *Nature*, 488, 353, 2012). For confirmation, UV/Vis absorbance spectra have been remeasured for the samples of interest, and stacked plots are shown here in Fig R1. The top spectra show no degradation of the pentacene peaks. At lower wavelengths, there is a slight reduction in peaks at 235 and 275 nm after several days each per sample under a 5mJ per pulse, 20Hz repetition rate laser excitation and over a year stored in ambient conditions. The more gradual tail towards higher wavelengths (blue lines) is attributed to scattering since data was remeasured without an integrating sphere. As these changes are also observed in the very dilute (0.5 %) sample, but no additional peaks are observed in EPR, it can be concluded the minor degradation does not affect the results and overall conclusion of the paper.

We can also conclude that there is no degradation of our samples prior to the optical characterisation, as the spectra below correspond to literature values in inert atmosphere, see e.g. ref. 35.

Fig R1. Absorption spectra of key samples, comparing the repetition, taken after ageing for a minimum of 6 months and denoted by “rep” compared to the original measurements labelled “orig”.

8. The spectral slices in Figure S9 do not have a time label.

All time labels have been added to spectral slices figures, including Figure S9.

9. In Table 1 the bipentacenes are conventionally labelled BP2 and BP3. I would also point the authors toward other pentacene dimer tr-ESR papers to fill out that table: Nature Communications volume 8, Article number: 15171 (2017); J. Am. Chem. Soc., 2017, 139 (36), pp 12488–12494 & Chem Volume 4, Issue 5, p1092–1111.

The bipentacenes have been relabelled to reflect the convention. We have also added the dimers suggested by Reviewer #3 with corresponding references.

Reviewer #2 (Remarks to the Author):

Reading carefully again the manuscript, I must confess I have been overly critical regarding the originality of the work. This was in part due to the writing style, now significantly improved in the revised manuscript. I am slowly but surely getting convinced by the authors that their EPR data allow distinguishing two different types of molecular packing. Fig.S6 is useful but I still think the conclusion at the end of the section 'orientation of pentacene molecules in the film', namely that the analysis reveals the presence of two different geometries, is not corroborated by the data at this point of the manuscript. To me, the correct conclusion is that there are different molecular orientations. It is only from the following section discussing the nature of the multiexciton spin states and the analysis of Fig.6 that the presence of different dimers can be inferred. Once the authors have fixed this minor issue, I think the paper would be suitable for publication.

Reviewer #3 (Remarks to the Author):

The authors have revised their manuscript considerably; they have made the work more clear and have added insights into the dynamics of their samples.

I believe that this is important work that furthers our understanding of the importance of intermolecular geometry on the dynamics of spin species. This work will provide theoreticians with excellent experimental data with which to understand their calculations. Finally, the surprising results that $S=2$ spin species exist at room temperature for many microseconds will guide the optical spectroscopy community when analysing long lived species.

I recommend this for publication in Nature Communications, subject to the two minor changes below.

1. In light of Reviewer #2's comments I suggest changing the instances of 'dimer' into 'aggregates' or 'clusters' in the abstract. Similar changes should be made throughout the manuscript, though I note that sometimes the word 'dimer' is appropriate.

2. Line 234 appears to have a notation error. As it's written it looks like the transitions are between spin multiplicities rather than m_S

Response to reviewers' comments:

Comments are reported in blue

Our response is reported in black

Reviewer #2 (Remarks to the Author):

Reading carefully again the manuscript, I must confess I have been overly critical regarding the originality of the work. This was in part due to the writing style, now significantly improved in the revised manuscript. I am slowly but surely getting convinced by the authors that their EPR data allow distinguishing two different types of molecular packing.

We are pleased that the reviewer appreciates the originality and conclusions of our paper.

Fig.S6 is useful but I still think the conclusion at the end of the section 'orientation of pentacene molecules in the film', namely that the analysis reveals the presence of two different geometries, is not corroborated by the data at this point of the manuscript. To me, the correct conclusion is that there are different molecular orientations. It is only from the following section discussing the nature of the multiexciton spin states and the analysis of Fig.6 that the presence of different dimers can be inferred. Once the authors have fixed this minor issue, I think the paper would be suitable for publication.

We have agree that even though our analysis in that section identified two pairs of triplets, their attribution to different intermolecular geometries is only confirmed in the following section.

We therefore added a "subsequent" in

*The appearance of the inner peaks correlates with structural and optical data, which demonstrates aggregation above this concentration, and our **subsequent** analysis shows these peaks belong to quintets formed by pentacene dimers with two distinct geometries, either parallel, ${}^5(TT)_{||}$, or herringbone, ${}^5(TT)_{\wedge}$...*

Reviewer #3 (Remarks to the Author):

We are very pleased that the reviewer values our work, and its impact in the context of future theoretical and spectroscopic studies.

1. In light of Reviewer #2's comments I suggest changing the instances of 'dimer' into 'aggregates' or 'clusters' in the abstract. Similar changes should be made throughout the manuscript, though I note that sometimes the word 'dimer' is appropriate.

We agree that clusters which contain more than two molecules will also be affected by the changes we identified and have therefore added "and aggregates" in the abstract.

*In this work, we produce **highly ordered dilute pentacene films with distinct parallel and herringbone dimers and aggregates***

2. Line 234 appears to have a notation error. As it's written it looks like the transitions are between spin multiplicities rather than m_S

We thank the reviewer for spotting this error and have now fixed it.